# Effects of Different Degrees of *Xanthium spinosum* Invasion on the Invasibility of Plant Communities in the Yili Grassland of Northwest China

**DOI:** 10.3390/biology13010014

**Published:** 2023-12-26

**Authors:** Yongkang Xiao, Jianxiao He, Tayierjiang Aishan, Xiaoqing Sui, Yifan Zhou, Amanula Yimingniyazi

**Affiliations:** 1College of Grassland Sciences, Xinjiang Agricultural University, Urumqi 830052, China; xiao_yk2023@163.com (Y.X.); xjau1952@163.com (J.H.); sxq303@xjau.edu.cn (X.S.); z_yf0930@163.com (Y.Z.); 2College of Ecology and Environment, Xinjiang University, Urumqi 830046, China; tayirjan@xju.edu.cn; 3Xinjiang Key Laboratory for Ecological Adaptation and Evolution of Extreme Environment Biology, College of Life Sciences, Xinjiang Agricultural University, Urumqi 830052, China; 4Institute of Plant Protection, Xinjiang Academy of Agricultural Sciences, Key Laboratory of Integrated Pest Management on Crops in Northwestern Oasis, Ministry of Agriculture, Xinjiang Key Laboratory of Agricultural Biosafety, Urumqi 830091, China

**Keywords:** invasive species, biodiversity, community stability, community invasibility

## Abstract

**Simple Summary:**

The effects of invasive plants on species diversity and ecosystem stability differ owing to the different stages of invasion. *Xanthium spinosum* is an invasive weed that is widely distributed in Europe, Asia, and North America. Its adaptability and competitive advantage over native species have led to agricultural loss and biodiversity reduction, yet its impact on local plant communities under different degrees of invasion has not been reported. Therefore, in this study, we aimed to investigate the effects of different degrees of invasion by *X. spinosum* on the diversity, stability, and invasibility of local plant communities in Yining County, Xinjiang, China. We believe that our study makes a significant contribution to the literature because we found that as the degree of *X. spinosum* invasion increased, species diversity and community stability decreased, while community invasibility increased. Notably, a light degree of invasion was found to enhance the diversity and stability of local plant communities. Based on our findings, we recommend that *X. spinosum* be controlled and eradicated in the early stages of invasion to prevent further invasion and harm to indigenous species.

**Abstract:**

Studying the effects of different degrees of exotic plant invasion on native plants’ community structure and plant diversity is essential for evaluating the harm caused to ecosystems by plant invasion. In this study, we investigated the effects of *Xanthium spinosum*, a widespread invasive species, on plant community species diversity and community stability in the Ili River Valley area of Xinjiang, China, under three invasion levels (no invasion and low, moderate, and heavy invasion), and the competitive advantage index, invasion intensity, and contribution of plant community species diversity to community stability and invasibility were determined for the prickly fungus under different degrees of invasion. The results show that there were significant differences (*p* < 0.05) in the species diversity and community stability of plant communities caused by different degrees of invasion of *X. spinosum*. The species diversity and stability of plant communities were negatively correlated with the community invasibility, competitive advantage, and invasion intensity of *X. spinosum* (*p* < 0.05); therefore, the competitive advantage and invasion intensity of *X. spinosum* increase with the increase of its invasion degree. On the contrary, community species diversity and stability decreased with the increase of its invasion degree, ultimately leading to differences in community invasibility under different invasion degrees. The Shannon–Wiener and Simpson’s indices were the greatest contributors to community stability and invasibility, respectively. Moderate and heavy levels of invasion by *X. spinosum* reduced the diversity and stability of local plant communities, increased the invasibility of communities, and substantially affected the structures of plant communities. Therefore, the continued invasion by *X. spinosum* will have an immeasurable impact on the fragile ecosystems and diversity of indigenous species in Xinjiang. We recommend that this invasive species be controlled and eradicated at the early stages of invasion to prevent further harm.

## 1. Introduction

The invasibility of plant communities refers to the degree to which they are vulnerable to invasion by alien plants [1,2,3]. The main factors that determine the invasibility of plant communities include plant diversity, characterized by genetic variability, species diversity, and functional diversity, and plant community stability, which depends on the ability of the community to maintain the combination of species and the quantitative relationships among species in a certain period and restore the original equilibrium state under disturbance [3,4,5]. Studying the invasibility of plant communities has theoretical and practical relevance for understanding the mechanisms of alien plant invasion, preventing and controlling invasion by alien plants, and restoring invaded communities [6,7]. Studies have investigated the invasive mechanisms of alien plants (e.g., biological characteristics and interspecific relationships) and the effects of climate or environmental changes on plant invasion, but few studies have focused on the relationship between plant invasion and community invasibility [8,9].

The effects of invasive plants on species diversity and ecosystem stability differ owing to the different stages of invasion (introduction, settlement, latency, dispersal, and outbreak) [10,11,12]. For example, the invasive plants *Ageratina adenophora* and *Eupatorium odoratum*, which invade multiple habitats in Guangxi Province, China, inhibit the growth of native plants through competitive advantages and allelopathy, seriously reducing species diversity and community stability [13]. Additionally, *Bidens pilosa* seriously damages the native plant community structure and reduces species diversity through its strong invasiveness, destroying the stability of the ecosystem [14]. These studies have shown that invasive plants can severely damage species diversity and ecosystem stability in invaded areas. However, some studies have shown that the species diversity of *Alternanthera philoxeroides* under low invasion is higher than that under severe invasion [15]. Thus, studying the impact of different degrees of invasive alien plants on the regional community structure and species diversity has important theoretical and practical significance in evaluating their harm to ecosystems. Existing studies have mainly divided the survey areas into invasive and non-invasive areas rather than clearly assessing the impact of different degrees of invasion on species diversity [16,17].

*Xanthium spinosum* is an annual invasive weed belonging to the Compositae family. This species is native to South America but is widely distributed in Central and Southern Europe, Asia, and North America, owing to its strong invasiveness [8]. In China, it was first discovered in Dancheng County, Henan Province, in 1981. Since then, *X. spinosum* has spread to Anhui Province, Beijing, Liaoning, Inner Mongolia, Ningxia, Xinjiang, and six other provinces [18]. This species displays strong adaptability, reproductive ability, and competitive ability in its invasiveness, which not only causes agricultural loss and biodiversity reduction but also affects the health of humans and livestock owing to pollen allergies [19]. The thorns on the surface of the involucre of *X. spinosum* allow it to be easily spread by humans or animals when it attaches to clothing or fur, which can inadvertently spread it over a large area within a short time [9]. In addition, *X. spinosum* has strong growth and reproductive abilities, a large seed yield, a high germination rate, diverse diffusion media, and rapid spread, and it quickly occupies a large area and inhibits the growth and reproduction of native plants and crops. Furthermore, this species is slightly toxic and has many sharp, yellow, hard spines on both sides of the petiole base [20]. When the spines pierce human or animal skin, it causes severe pain for an extended time, which can interfere with cattle and sheep husbandry [18,21]. Therefore, *X. spinosum* has been included in the updated alien invasive species list of natural ecosystems in China and the list of key management of invasive species [22].

Studies on the effects of *X. spinosum* on local plant communities have only discussed the substantial effects on plant community diversity from the perspective of changes in the quantitative characteristics of plant communities in the invaded area, namely height, coverage, density, and importance [21]; however, the effect of *X. spinosum* on the invasibility of local plant communities under different degrees of invasion has not been reported. Therefore, because of the wide distribution of and damage caused by *X. spinosum* in Yining County, Xinjiang, China, we aimed to investigate the effects of different degrees of invasion by *X. spinosum* on the diversity, stability, and invasibility of local plant communities.

The main objectives of the present study were to (1) evaluate the effects of different degrees of *X. spinosum* invasion on species diversity, community stability, and community invasibility and identify the influencing factors, (2) determine the relationships among species diversity, community stability, and community invasibility, and (3) identify which species diversity index contributes the most to community stability and the community stability index. The results of this study provide a theoretical basis for the scientific management of *X. spinosum* and local biodiversity protection.

## 2. Materials and Methods

### 2.1. Study Site

The sampling area of this study was located in Wugong Township (81°35′ E, 44°17′ N; elevation 860 m), Yili Prefecture, Xinjiang. The habitat type was desert steppe, and the survey area was approximately 2.1 ha. The sampling area has a temperate continental semi-humid desert climate. The annual average temperature is approximately 9 °C, and the monthly average temperature is as high as 32 °C in July and as low as −11 °C in January. The annual precipitation is approximately 340 mm, the annual sunshine duration is approximately 2900 h, and the average annual evaporation is 1621 mm. The plant community in the sampling area comprised erbaceous weeds. Samples of *X. spinosum* and coexisting native plants in the sampling area were investigated, and the only invasive plants in the community were *X. spinosum*.

### 2.2. Experimental Design

Referring to the classification method used by previous researchers and taking into account the actual situation in the sampling area, the degree of invasion of *X. spinosum* was evaluated based on its population coverage in the invaded area [12]. Four levels of invasion were designated: no invasion (0%, CK), low invasion (<35%, L), moderate invasion (35–75%, M), and heavy invasion (>75%, H). Based on the estimated distribution of *X. spinosum*, 40 quadrats (2 m × 2 m) were investigated in September 2023, and all herbaceous species in the quadrats were evaluated. The number and height of each plant species in the quadrats were measured, and coverage was estimated. Three individual plants with vigorous growth were selected from each species to determine plant functional traits, and three mature and complete leaves were randomly selected to determine leaf functional traits.

### 2.3. Determination of Functional Traits for X. spinosum

For the determination of functional traits, we selected 10 functional traits closely related to the growth, competitiveness, and fitness of the invasive and native plants [12,23]. The plant height, leaf length, leaf width, and petiole length were measured using a ruler with an accuracy of 0.1 cm. The ground diameter, petiole diameter, and leaf thickness of the plants were measured using a Vernier caliper with an accuracy of 0.01 mm. A plant nutrition analyzer (TYS-3N; Fansheng Technology Co., Ltd., Shijiazhuang, China) was used to determine the chlorophyll and leaf nitrogen of plants, and a leaf image analyzer (FS-leaf1000; Fansheng Technology Co., Ltd., Shijiazhuang, China) was used to measure the leaf area of plant leaves.

### 2.4. Determination of the Correlation Index for X. spinosum

In this study, α diversity was used to reflect the level of species organization in the community. The specific indices used were as follows: Shannon–Wiener (H, plant diversity) [24], Simpson’s (D, plant dominance) [25], Pielou’s (J, community evenness) [26], and Margalef’s (F, plant richness) [27].

To quantify the ecological impact risk of *X. spinosum* on plant species diversity and community stability, we used the following six degree index indicators (DIIs): the influence degree indices of (1) *X. spinosum* on species number (S) (DII_S_), (2) *X. spinosum* on the Shannon–Wiener index (DII_H_), (3) *X. spinosum* on Simpson’s index (DII_D_), (4) *X. spinosum* on Pielou’s index (DII_J_), (5) *X. spinosum* on Margalef’s index (DII_F_), and (6) *X. spinosum* on community stability (DII_ICV_). The specific calculation formula and detailed methodology are in Wang et al. [12].

The community stability index (ICV) and community invasibility index (CII) were calculated according to the methods of Wang et al. [28].

The competitive advantage index (CAI) was used to characterize the competitive advantage of *X. spinosum* under different degrees of invasion. The invasion intensity of *X. spinosum* under different degrees of invasion was characterized using the invasion intensity index (III). The specific calculation formulae and methodology are in Wang et al. [12,28].

### 2.5. Statistical Analyses

Excel 2019 was used for data sorting, and SPSS 25.0 was used to analyze the indicators. The explore command was used to test the normality of the data, while Tukey’s multiple comparison tests were used to determine the differences between the average values of each data group. A one-way ANOVA analysis of variance was used to compare the differences in plant species diversity, stability, and invasion ability under different levels of invasion, with a significance level set at *p* < 0.05. The Pearson product moment correlation coefficient was used for correlation analysis to determine the relationships between community diversity, community stability, and the invasibility of plant communities invaded by *X. spinosum* to different degrees. We also analyzed the contribution of community species diversity to community stability and invasibility through the direct path coefficient (P) and indirect path coefficient (P’) of path analysis. Mapping was completed using Origin 2021.

## 3. Results

We found that different degrees of invasion by *X. spinosum* significantly affected the diversity index of the local plant communities (*p* < 0.05). As the degree of invasion increased, the diversity index of the local plant communities first increased and then decreased. The H’ and D indices were significantly different at the four invasion levels (*p* < 0.05), and the maximum values were 1.41 and 1.29, respectively, under low invasion. Pielou’s and Margalef’s indices were higher under the low invasion level (0.82 and 1.04, respectively) than under the other three levels, and they were significantly higher under the no invasion and moderate invasion levels than under the heavy invasion level (*p* < 0.05). This shows that low invasion by *X. spinosum* increases the diversity of local plant communities and that heavy invasion by *X. spinosum* reduces the diversity of local plant communities (Figure 1).

The ICV decreased with increasing degrees of invasion. Compared with the ICV under low invasion, there was a significant decrease of 30% under heavy invasion (*p* < 0.05). By contrast, with an increase in the degree of invasion, the CII, CAI, and III showed an upward trend. Additionally, compared with the indices under low invasion, there were significant increases of 88%, 92%, and 89%, respectively, in the three indices under heavy invasion (*p* < 0.05). This shows that the dominance and invasion intensity of *X. spinosum* and the invasibility of the plant community increased with an increase in the invasion degree of *X. spinosum* but that the stability of the community decreased (Figure 2).

The impact index of *X. spinosum* on the diversity of local plant communities varied with different invasion degrees. Compared with that of low invasion by *X. spinosum*, the impact index of *X. spinosum* on the local plant community diversity under moderate and heavy invasion was positive (i.e., moderate and heavy invasion suppressed the diversity index of local plant communities). During low invasion, the impact index of *X. spinosum* on the diversity of local plant communities was negative (i.e., low invasion promoted the diversity index of local plant communities), with the greatest impact on Simpson’s index being −0.57. Low invasion by *X. spinosum* increased the diversity of local plant communities, especially Simpson’s index, and moderate and heavy invasion reduced the diversity of local plant communities (*p* < 0.05, Figure 3).

As the degree of invasion by *X. spinosum* increased, the impact index of *X. spinosum* on the ICV and S showed an upward trend. Under low and moderate invasion levels, the DII_ICV_ was <0, but under heavy invasion levels, the DII_ICV_ was >0. The DII_s_ was <0 for low invasion and >0 for moderate and heavy invasion, and there was a significant difference between different invasion degrees (*p* < 0.05). As the degree of invasion by *X. spinosum* increased, it damaged the stability of the local community and reduced the number of community species (Figure 4).

The correlation analysis of the local plant community diversity index with community stability and invasibility showed a highly significant positive correlation between H’, D, and J of community diversity and ICV (*p* < 0.001), and the maximum positive correlation coefficient between J and ICV was 0.831. However, the correlations between S, and F, and ICV were not significant. The S, H’, D, J, and F of community diversity were significantly negatively correlated with CII, CAI, and III (*p* < 0.001), and the negative correlation coefficient between D and CII was the smallest at −0.99. This indicates that community diversity was positively correlated with community stability and negatively correlated with community invasibility (Figure 5).

According to the path analysis results of species diversity’s impact on community stability and invasibility, the magnitude of the contribution of each diversity index to community stability was in the following order: H’ > F > S > D > J. Therefore, plant diversity had the greatest contribution to community stability (*p* = 2.264). Additionally, H had a significant indirect contribution to community stability through D (P’ = 2.218). The contribution of each diversity index to community invasibility was in the order of D > H’ > F > S > J. Therefore, dominance had the greatest contribution to community invasibility (*p* = 0.579), and dominance had a significant indirect contribution to community invasibility through its influence on plant diversity (P’ = 0.565, Table 1).

## 4. Discussion

Invasion by non-native plants has a substantial impact on local species diversity, community stability, and community invasibility [12,19]. In this study, the impact index of low invasion by *X. spinosum* on local species diversity, community stability, and community invasibility was negative, indicating a significant increase in local species diversity compared to that in uninvaded areas. Other studies have reached similar conclusions [29,30,31]. The reasons for these results may be that the competitive ability or allelopathy of invasive plants in the early stages of invasion is insufficient to cause a large impact and that crowding out of local species is not yet apparent. Instead, communities under low invasion exhibit species diversity similar to that of uninvaded communities, demonstrating a positive impact on community species diversity [28]. By contrast, in this study, a heavy level of invasion significantly reduced local species diversity and community stability while increasing community invasibility. This is likely due to the fact that as the degree of invasion by *X. spinosum* increased, its invasion intensity and competitive advantage continued to increase, resulting in a positive impact index of heavy invasion by *X. spinosum* on local species diversity, community stability, and community invasibility. This led to the loss of local plant diversity and serious damage to community stability, resulting in negative impacts and ultimately increasing community invasibility.

A few studies have demonstrated that prickly ash fungus had a stronger survival ability than the local plants, creating favorable conditions for its invasion [32,33]. Thus, invasion by alien plants does not necessarily lead to a decrease in the species diversity and community stability of local communities, leading to an increase in invasibility. There were significant differences in the impact of *X. spinosum* on community species diversity at different invasion levels [34,35]. Low invasion by alien plants can increase species diversity in plant communities, improve community stability, fill empty niches, utilize available resources, and reduce community invasibility [36].

There is a close relationship between the invasion and stability of the community and the species diversity of the local plant community [1,37,38]. The present study showed that community invasibility was negatively correlated with species diversity and community stability, indicating that species diversity and community stability decreased with an increase in the invasion degree of *X. spinosum*. However, community invasibility increased, and the competitive advantage of *X. spinosum* increased its coverage and was negatively correlated with species diversity and community stability. The level of invasion by *X. spinosum* increased, and its strong competitive advantage disrupted the balance of the local ecosystem, leading to a decrease in local plant diversity and community stability, and under heavy levels of invasion, the extinction of other plants may occur [38,39]. However, there are inconsistencies in the literature. For example, invasive plants have been shown to have higher community invasiveness in ecosystems with higher species diversity, showing a positive correlation [40,41]. Additionally, the invasibility of the community has been shown to be not significantly related to the species diversity of local plant communities [31,42]. Therefore, the relationship between community invasibility, species diversity, and plant community stability is relatively complex [12].

The differences between local plant communities (for example with regard to aspects such as structure, size, and stability) can lead to changes in the species diversity and invasive relationships of different communities. Species diversity, as the core component of plant communities, affects the stability and invasibility of communities [36,40,43]. This study showed that the contribution of H’ to community stability was higher than that of other diversity indices, indicating that communities with higher diversity are likely to have higher levels of community stability and develop greater resistance to plant invasion. A previous study reached similar conclusions [12]. However, another previous study suggested that plant dominance is crucial for maintaining community stability [43]. In addition, the contribution of D to community invasibility was higher than that of the other diversity indices, which is inconsistent with results in the literature [12]. As the CAI and III of *X. spinosum* increased, it became the dominant species in the invaded community, significantly affecting the invasibility of the community. Thus, diversity plays a crucial role in the stability of communities, and dominance has a notable impact on the invasibility of communities.

## 5. Conclusions

Different degrees of invasion by *X. spinosum* changed the species diversity, community stability, and community invasibility of local plant communities. The degree of the impact of *X. spinosum* on the species diversity and stability of local plant communities increased with the degree of invasion. The degree of impact index ranged from <0 for low invasion to >0 for heavy invasion. During this period, the species diversity and stability of the community decreased significantly, and the invasibility of the community increased significantly. The competitive advantage of *X. spinosum* was negatively correlated with community species diversity and community stability but positively correlated with community invasibility. In addition, community invasibility was negatively correlated with community species diversity and stability. Therefore, the relationship between the invasibility of plant communities and the species diversity and stability of plant communities indicates that plant communities with lower invasibility and higher species diversity and stability of plant communities are more resistant to invasive plants. Among the communities invaded by *X. spinosum*, diversity had the greatest contribution to community stability, and dominance had the greatest contribution to community invasibility. Moderate and heavy invasion by *X. spinosum* reduced the diversity and stability of local plant communities, increased the invasibility of communities, and significantly affected the structure of plant communities. Therefore, invasion by *X. spinosum* will likely have an immeasurable impact on the fragile ecosystems and diversity of indigenous species in Xinjiang. *X. spinosum* should be controlled and eradicated in the early stages of invasion to prevent pernicious invasion and further harm by this invasive species.

## Figures and Tables

**Figure 1 biology-13-00014-f001:**
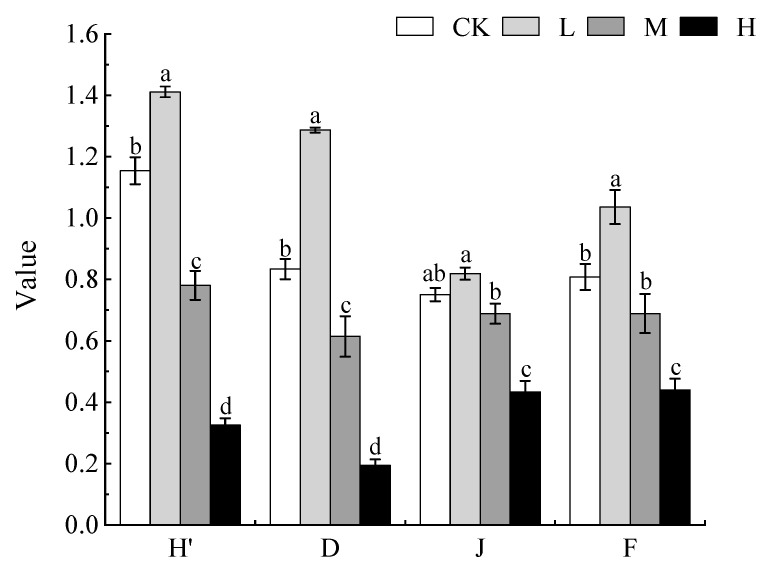
The variation trend of the local plant community diversity index under different degrees of *Xanthium spinosum* invasion. CK represents no invasion of *X. spinosum*; L represents low invasion of *X. spinosum*; M represents moderate invasion of *X. spinosum*; H represents heavy invasion of *X. spinosum*; H’ represents the Shannon–Wiener index; D represents Simpson’s index; J represents Pielou’s index; F represents Margalef’s index. Different lowercase letters indicate significant differences in species diversity indices under different levels of invasion by *X. spinosum* (mean ± SE, *p* < 0.05).

**Figure 2 biology-13-00014-f002:**
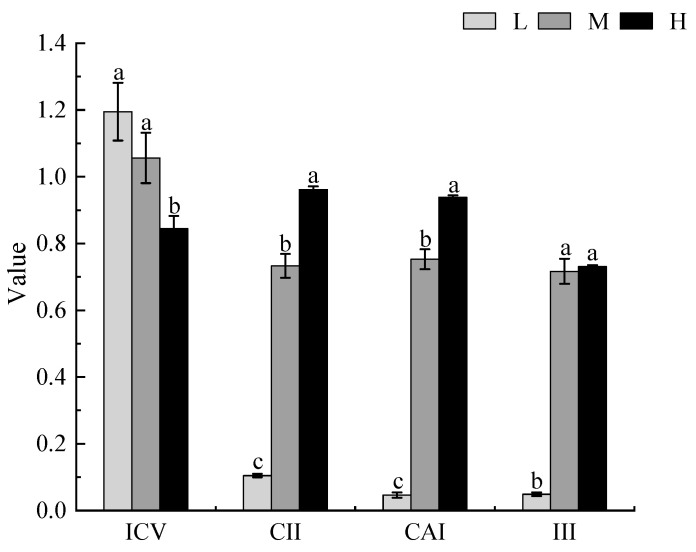
The changes in the stability, invasiveness, competitive advantage index, and invasion intensity index of the local plant community under different degrees of *Xanthium spinosum* invasion. L represents low invasion of *X. spinosum*; M represents moderate invasion of *X. spinosum*; H represents heavy invasion of *X. spinosum*; ICV represents the community stability index; CII represents the community invasiveness index; CAI represents the competitive advantage index of *X. spinosum*; III represents the invasion intensity index of *X. spinosum*. Different lowercase letters indicate significant differences under different degrees of invasion by *X. spinosum* (mean ± SE, *p* < 0.05).

**Figure 3 biology-13-00014-f003:**
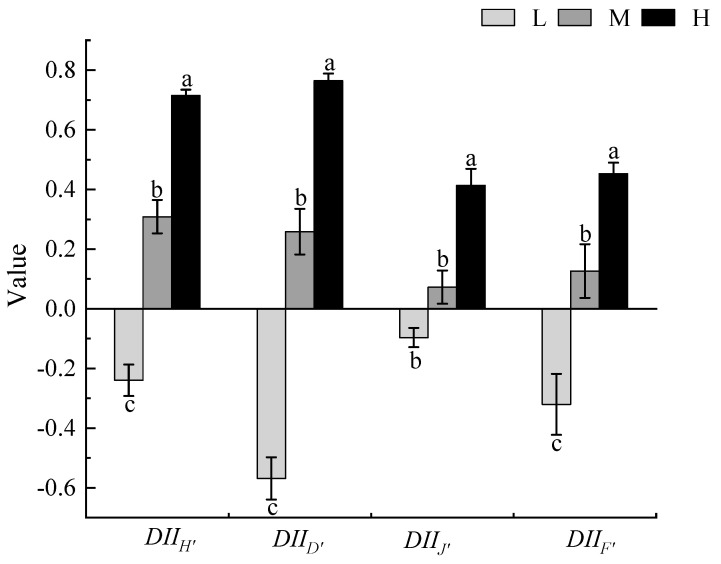
The influence of different degrees of *Xanthium spinosum* invasion on the diversity index of local plant communities. L represents low invasion of *X. spinosum*; M represents moderate invasion of *X. spinosum*; H represents heavy invasion of *X. spinosum*; DII_H’_ represents the index of the degree of influence of *X. spinosum* on the Shannon−Wiener index; DII_D’_ represents the index of the degree of influence of *X. spinosum* on Simpson’s index; DII_J’_ represents the index of the degree of influence of *X. spinosum* on Pielou’s index; DII_F’_ represents the index of the degree of influence of *X. spinosum* on the Margalef’s index. Different lowercase letters indicate significant differences under different degrees of invasion by *X. spinosum* (mean ± SE, *p* < 0.05).

**Figure 4 biology-13-00014-f004:**
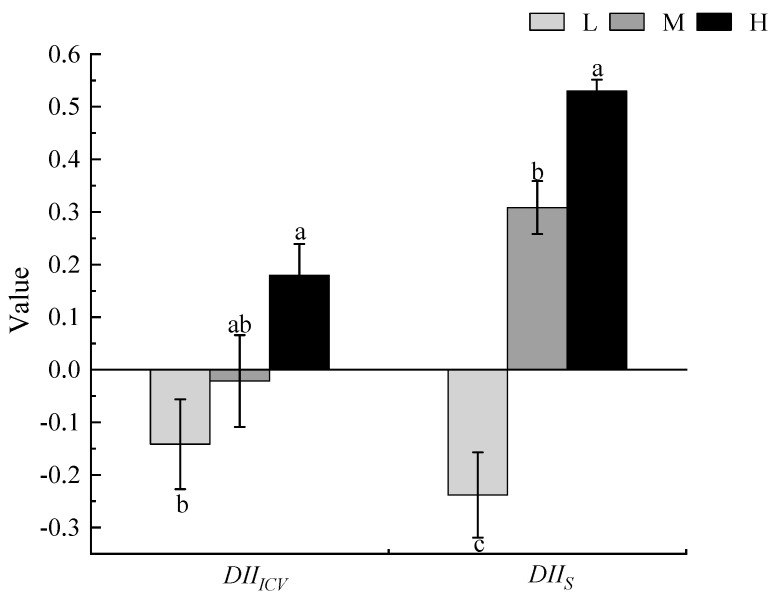
The influence of different degrees of *Xanthium spinosum* invasion on the species number and stability index of local plant communities. L represents low invasion of *X. spinosum*; M represents moderate invasion of *X. spinosum*; H represents heavy invasion of *X. spinosum*; DII_ICV_ represents the degree index of the impact of *X. spinosum* on the community stability index; DII_S_ represents the index of the degree of influence of *X. spinosum* on the number of plant species in the community. Different lowercase letters indicate significant differences in the degree of invasion of *X. spinosum* (mean ± SE, *p* < 0.05).

**Figure 5 biology-13-00014-f005:**
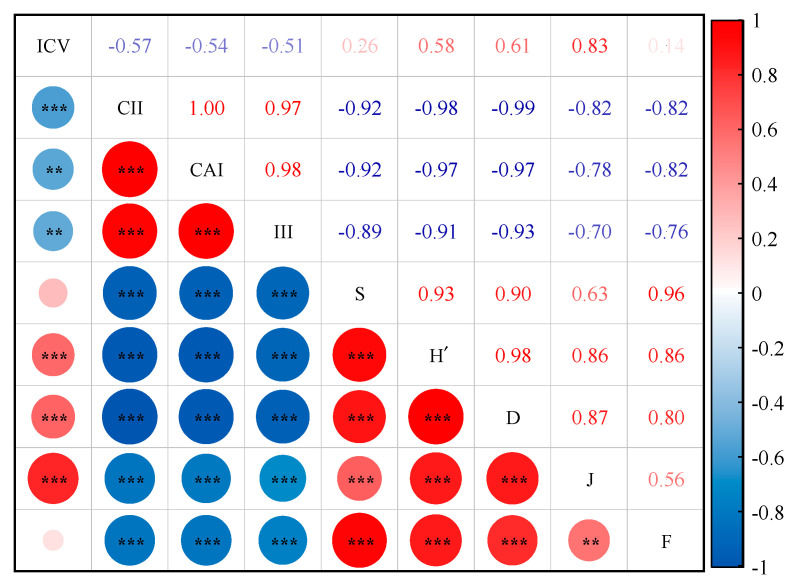
The correlation between the local plant community diversity index and community stability and invasibility. ICV is the community stability index; CII is the invasiveness index; CAI is the competitive advantage index of *X. spinosum*; III is the invasion intensity index of *X. spinosum*; S is the number of plant species in the community; H’ is the Shannon−Wiener index; D is Simpson’s index; J is Pielou’s index; F is Margalef’s index; **, *p* < 0.01; ***, *p* < 0.001.

**Table 1 biology-13-00014-t001:** The contribution of plant community diversity to community stability and invasibility.

Community Stability Index							Community Invasibility Index					
	*P*	*P’*					*P*	*P’*				
		S	H’	D	J	F		S	H’	D	J	F
S	−0.819	-	0.761	0.737	0.515	0.786	−0.323	-	0.299	0.290	0.204	0.309
H	2.264	2.106	-	2.218	1.947	1.947	−0.569	0.527	-	0.559	0.491	0.491
D	−0.350	0.315	0.343	-	0.304	0.280	−0.575	0.516	0.565	-	0.501	0.463
J	0.176	0.110	0.151	0.153	-	0.098	0.184	0.116	0.159	0.160	-	0.103
F	−0.848	0.814	0.729	0.678	0.474	-	0.344	0.330	0.297	0.277	0.192	-

P is the direct path coefficient; P’ is the indirect path coefficient; S is the number of plant species in the community; H’ is the Shannon–Wiener index; D is Simpson’s index; J is Pielou’s index; F is Margalef’s index.

## Data Availability

The original contributions presented in the study are included in the article; further inquiries can be directed to the corresponding author.

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
