# Peer review of "Effects of Different Degrees of Xanthium spinosum Invasion on the Invasibility of Plant Communities in the Yili Grassland of Northwest China"

_biology, 2023, doi:10.3390/biology13010014_

Round 1

Reviewer 1 Report

Comments and Suggestions for Authors

The section of Introduction is written quite well. The main background and the relevance of the study are highlighted there. I think that this section can be published as it is.

In Material and Methods, please, add the map of the study area with inclusion the scheme of the arrangement of study plots. 

The section Results looks excellent. All graph plots and tables clearly describe the obtained results. This section clearly informs us on the significant influence of an invasive plant species on the natural plant communities. It is especially great, since the authors used a set of functional traits of the plants.

The section Discussion should be revised. For instance, it includes the description of the obtained results (e.g. see lines 265-275). At the same time, I recommend to exclude such irrelevant data from the Discussion. This section should include the explaining and discussing the obtained data in light of the modern literature and other relevant sources on influence of biological invasions on ecosystems and populations of organisms. However, the section is generally written quite well, and no crucial problems are seen.

Finally, the section Conclusions is well written, and I have no significant requests.

Reviewer 2 Report

Comments and Suggestions for Authors

I have critically evaluated the manuscript by Xiao et al. submitted to Biology. Authors have studied the impacts of varying invasion levels on the structure and stability of grassland communities in a province China. The submitted work is meticulously planned and organised nicely. I have few minor suggestions to improve presentation and clarity of the work.

1. In the abstract, pl remove redundancies, see lines 16-21; seems repetitive. Better to explain the quantitative differences in community structure due to varying degree of invasion, rather than just mentioning which was more detrimental.

In fact, abstract be revised and rewritten to highlight what has been studied; authors have studied functional traits but no mention of anything like that is there! Abstract should be standalone.

2. In lines 91-95, include information related to what actually has been investigated, i.e. parameters studied and why?

3. line 100, instead of giving coordinates of the province/study, better give a range of these coordinates in the study site/area.

Provide a map of the study site

Revise lines 102-108 to provide only relevant information and all this be detailed under study site, and not under experimental design, which is altogether a different aspect!

4. What was the rationale for the choice of functional traits studied.

5. Provide details of the other herbaceous flora found in the study site. How were invasion classes defined? Provide more details.

6. Pl revise statistical analyses section. How can we get mean+SD from ANOVA? 

What is meant by indicators?

What mapping was completed? Where are the details?

Authors need to strengthen the discussion in view of available literature. Pl see the papers by research groups of Batish/Singh/ Ahmand, published in Ecological Indicators, STOTEN, Journal of Environmental Management, etc.

Comments on the Quality of English Language

At few places, authors need to pay attention for the grammar etc.

Reviewer 3 Report

Comments and Suggestions for Authors

The study titled "Impacts of Varying Degrees of Xanthium spinosum Invasion on Grassland Plant Communities in the Yili River Valley, Xinjiang, China" delves into the ramifications of Xanthium spinosum, an extensively invasive species, on the species diversity and stability of plant communities within the Ili River Valley region of Xinjiang, China. The research encompasses three invasion levels—absence of invasion, low invasion, moderate invasion, and heavy invasion—providing comprehensive insights into the varying degrees of invasibility.

The outcomes of the investigation underscore that moderate and heavy levels of X. spinosum invasion detrimentally affect the diversity and stability of local plant communities. Additionally, these invasive levels intensify the susceptibility of communities to further invasions, significantly altering the overall structure of plant communities. The authors advocate for the early-stage control and eradication of this invasive species to mitigate potential harm.

Commendably, the title aptly encapsulates the study's focus, and the authors list is recommended to be checked for corresponding author details. The abstract is well-articulated, with a suggestion for clarifying the description of invasion levels in line 15.

In the introduction, while the identification of the research gap is well-executed, certain sentences, particularly in lines 34-39, are overly lengthy and necessitate revision for improved clarity. Additionally, it is suggested to present author names with the corresponding scientific names upon first mention, addressing all binomials consistently. The integration of more up-to-date citations could enhance the manuscript's relevance for readers.

The methodology section is commendable, with a specific suggestion to include a map of the study area for improved manuscript quality. Attention is drawn to line 141 for citation format consistency.

The results section is well-structured, accompanied by clear and self-explanatory figures. However, there is a recommendation to consistently use 'H'' for the Shannon-Wiener index, replacing 'H' where necessary.

The conclusion is deemed appropriate, summarizing the key findings effectively. In the references section, concerns are raised regarding the use of references older than five years, prompting a request for updating some of these citations. Additionally, there is a query about the intentional use of multiple citations from the journal "Oikos."

 Overall, the study is acknowledged for its commendable efforts, with constructive feedback provided to enhance clarity, consistency, and relevance throughout the manuscript.
